# Deep Feature Meta-Learners Ensemble Models for COVID-19 CT Scan Classification

Jibin B. Thomas [1], Shihabudheen K. V. [1], Sheik Mohammed Sulthan [2] and Adel Al-Jumaily [2,3,4,5,*]

1 Department of Electrical Engineering, National Institute of Technology, Calicut 673601, India
2 Faculty of Engineering, Universiti Teknologi Brunei, Bandar Seri Begawan BE1410, Brunei
3 ENSTA Bretagne, French State Graduate, 29200 Brest, France
4 School of Computing Mathematics and Engineering, Charles Sturt University, Wagga Wagga, NSW 2795, Australia
5 School of Science, Edith Cowan University, Joondalup, WA 6027, Australia
* Correspondence: adel.al-jumaily@ieee.org

**Abstract:** The infectious nature of the COVID-19 virus demands rapid detection to quarantine the infected to isolate the spread or provide the necessary treatment if required. Analysis of COVID-19-infected chest Computed Tomography Scans (CT scans) have been shown to be successful in detecting the disease, making them essential in radiology assessment and screening of infected patients. Single-model Deep CNN models have been used to extract complex information pertaining to the CT scan images, allowing for in-depth analysis and thereby aiding in the diagnosis of the infection by automatically classifying the chest CT scan images as infected or non-infected. The feature maps obtained from the final convolution layer of the Deep CNN models contain complex and positional encoding of the images' features. The ensemble modeling of these Deep CNN models has been proved to improve the classification performance, when compared to a single model, by lowering the generalization error, as the ensemble can meta-learn from a broader set of independent features. This paper presents Deep Ensemble Learning models to synergize Deep CNN models by combining these feature maps to create deep feature vectors or deep feature maps that are then trained on meta shallow and deep learners to improve the classification. This paper also proposes a novel Attentive Ensemble Model that utilizes an attention mechanism to focus on significant feature embeddings while learning the Ensemble feature vector. The proposed Attentive Ensemble model provided better generalization, outperforming Deep CNN models and conventional Ensemble learning techniques, as well as Shallow and Deep meta-learning Ensemble CNNs models. Radiologists can use the presented automatic Ensemble classification models to assist identify infected chest CT scans and save lives.

**Keywords:** deep learning; COVID-19 prediction; ensemble classification; CT scan images

## 1. Introduction

Coronavirus disease (COVID-19) is an infectious disease caused by severe acute respiratory syndrome coronavirus 2 (SARS-CoV-2). The surge of the disease in late 2019 evolved into a pandemic which has affected billions of people and thousands of sectors. In addition to that COVID-19 has created a huge impact on physical, mental health and wellbeing of the people [1]. The reverse transcriptase-polymerase chain reaction (RT-PCR) test is regarded as the gold standard for confirming COVID-19 infection [2]. The shortage of RT-PCR kits and the need for intimate interaction with patients while testing can pose a major problem for rapid detection and infection discovery. Radiological analysis of Computed Tomography (CT) scans and X-ray chest images have proved effective in finding symptoms caused by the COVID-19 disease in infected patients [3] and can be formed as a first line of instantaneous detection before RT-PCR testing. The image analysis and accurate diagnosis required automatic tools with minimum human interruption, such as machine learning and deep learning. Deep learning has been gaining popularity in recent years

for analysis of unstructured data such as medical images, with deep convolutional neural networks (CNNs) being used extensively [4] due to their ability to capture complex and positional features, thereby aiding doctors while diagnosing patients.

Many of these state-of-the-art (SOTA) Deep CNN architectures are developed and open sourced by universities, companies, and research institutions for further improvements, thus contributing to the extraordinary pace of machine learning research and development. For domain-specific problems, these models may be retrained or fine-tuned with pre-trained weights available for popular or domain-specific databases. Various deep-learning models have been suggested and evaluated for COVID-19 prediction from CT scans images. A CNN based long short-term memory network is used for COVID-19 classification from pneumonia [5]. Better accuracy was obtained with the proposed model compared to SOTA machine learning models. A Genetic Deep Learning Convolutional Neural Network (GDCNN) was applied for effective COVID-19 prediction [6]. The estimation of COVID-19 diagnosis is performed using 3D CT images, which use different levels of ResNet architecture [7]. GoogleNet Inception v3 with transfer learning capability was proposed for efficient COVID prediction in [8]. Severity prediction for COVID-19 has been implemented using image segmentation and VB-Net [9]. Several efficient individual models have been reported on CT scan images for COVID-19 detection [10–13]. It has been shown that the detection of the most effective model from a group of CNN methods is ambiguous because the task involves several performance metrics. Various proven Deep CNNs such as AlexNet, VGG, DenseNet, SqueezeNet, GoogleNet (Inception Net), MobileNet, ResNet, Xception were used in experiments based on COVID radiology studies [14–16]. Efficient Nets were used as base feature extractors in [17–19]. These models proved to be useful with transfer learning, as they were pre-trained on the ImageNet database. Various improvements to the proven Deep CNNs were presented in [20–23] proposed spiking neural networks, which provided a F1 score of 0.74 for the CT scan database. These studies utilize a single CNN classifier which can overfit the test database, resulting in high variance and low generalization on real-world systems.

It is proved that ensemble learning of different CNN architectures in a suitable format will improve the feature extraction capability of machine learning [24]. Studies have shown that Ensemble modeling of the SOTA Deep CNN architectures produces better generalization on medical image databases compared to a single Deep CNN model [25–28] where Conventional Ensemble techniques in machine learning and Deep CNN make use of the base models' final predictions to improve the classification. Model voting techniques are already used to perform either average or majority votes to get final predictions. The other work to improve COVID-19 prediction used ensemble learning methods and Bagging techniques to train the base model on different instances of the training data and aggregate the model instances. Boosting techniques aggregate different instances of the model in which each model instance is an improved version of the previous model [29]. In this paper, we take advantage of the feature maps generated by Deep CNN architectures rather than the final predictions produced by the model. These deep feature maps are concatenated and fed into machine learning or deep meta-learners. The intuition behind this approach is that the deep feature maps characterize generic information about the data, allowing the meta-learners to extract better features pertaining to the data. The authors of [30] presented an adaptive boosting technique to improve the ResNet model for CT scan inference. The authors of [31] studied a gradient boosting technique over wide ResNet and compared its results to ResNet, DenseNet, and InceptionNet. The boosted Ensemble gave an accuracy of 87.7% for CT scan inference. The authors of [32] presented a multi stacked Ensemble in which the output of three meta learners, namely SVM, Autoencoder, and Naive Bayes, were combined using model voting for the final output. The authors of [33] presented a voting Ensemble of three base CNNs where majority voting of the final probabilities provided the final Ensemble output. [34] studied stacking generalization ensemble of the VGG where the database was bootstrapped to improve the VGG classifier's accuracy to 93.57%.

Most of the existing Ensemble techniques include voting, stacking, boosting, and/or Bagging the final predictions of the base learners. The Ensemble study is also limited to two or three CNNs as base learners and concentrates on binary classification, which results in less variance in the dataset. These models use Deep CNN classifiers as the base learners where the output feature maps from the Deep CNNs are passed on to a classification head to obtain the final probabilities. However, the feature maps represent deep spatial information pertaining to the CT scan image, which is lost when flattened for the classification head. Therefore, the feature maps offer a richer embedding compared to the final probabilities. This makes the feature maps more suited to be the input of the meta-learner while Ensemble modeling. The presented Ensemble models use the feature maps from four Deep CNN feature extractors and pass them to shallow and deep meta-learners for better generalization. The evaluation can be improved by using N-fold cross validation, thereby utilizing the entire database.

Deep learning approaches have been found to outperform learning machine algorithms while processing large databases with vast input features. With the introduction of attention mechanisms in [35], recent architectures such as Transformers have extensively been used for state-of-the-art (SOTA) published results [36,37]. Attention allows the model to focus on other embeddings in the ensemble that need to be paid attention to while processing one particular embedding, which allows for a more direct dependence between the embeddings of the Ensemble. Convolutional Neural Networks have been traditionally used to extract deep features from image datasets [38] and have been extended to other machine learning tasks due to their ability to capture complex and positionally invariant feature maps. From visualizations of the final convolutional activation representations of various Deep CNN architectures, it was observed that these models learn similar complex and positional features pertaining to the data [39,40]. Hence, it would be reasonable to learn the inter-dependence between the Ensemble feature maps through attention.

This paper contributes to the development of different ensemble learning mechanisms for machine learning and CNN to extract deep features from the chest CT scan image by utilizing the concept of deep meta-learning and attention mechanism. Initially Deep CNN architectures—namely VGG-19, Inception-V3, ResNet-152-v2, DenseNet-201, InceptionResNet-v2, Xception and Efficientnet-B7—that trained with five-fold cross-validation to extract deep features from the chest CT scan images. These classification models were optimized for the dataset by tuning their hyperparameters and were then evaluated on the test image database. Of these, the four best-performing models (ResNet-152-v2, Densenet-201, Xception, and EfficientNet-B7) have proceeded for ensemble learning. This paper proposes four groups of Ensemble learning mechanisms to predict COVID-19 from CT scan images.

i.     Normal machine learning meta-learning Ensemble CNNs models where the feature vectors extracted by the deep CNNs are concatenated and fed to various machine-learning models to classify the infected CT scan images.

ii.    Deep meta-learning Ensemble CNNs models that concatenate the extracted feature maps from the trained CNN models and then use Neural Network (NN) or Fully Convolutional Network (FCN) architectures to classify the infected CT scan images.

iii.   A novel Attentive Ensemble CNNs model that uses attention encoders to dynamically learn the relative importance between the feature maps extracted by the deep CNNs to classify the infected CT scan images.

iv.    Voting Ensemble Learning models that classify CT scan images by average voting or majority voting of the predicted probabilities of the four trained CNN models.

The rest of the paper is organized as follows. Section 2 provides a background of different deep CNN architectures, followed by brief descriptions of various machine learning and deep learning models used as meta-learners for ensemble modeling. Section 3 explains the methodology of existing and proposed ensemble learning approaches. Experimental evaluation and results are presented in Section 4, and finally, Section 5 concludes the paper.

## 2. Materials and Methods

### 2.1. Database

This study utilized the dataset of COVID-19, Normal, and Pneumonia chest CT scan axial slices from [41–46] integrated in [47] after removing duplicated datasets. The dataset has been used in COVID-19 diagnosis literature and has proven its effectiveness and efficiency in deep learning applications.

The entire database contains infected chest CT scan images of 7593 COVID-19 cases and 2618 viral pneumonia cases, as well as 6893 normal cases, as shown in Figure 1.

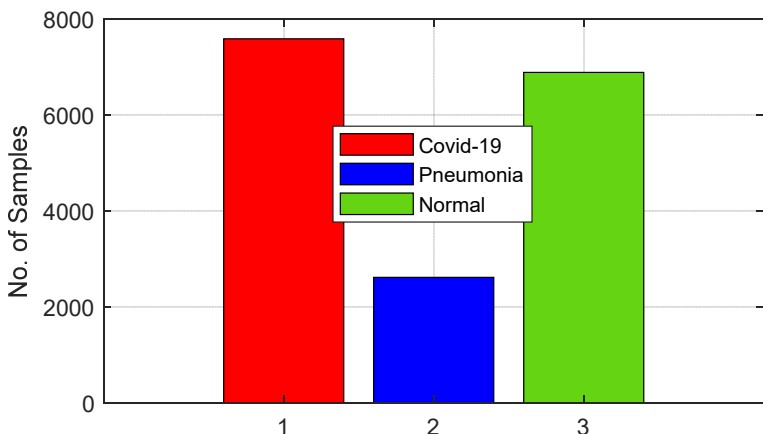

**Figure 1.** Distribution of Classes in the Image Database.

Figure 2 provides some sample lung CT scan images of COVID-19 infected patients. Figures 3 and 4 show some lung CT scan image samples of viral pneumonia patients and some normal lung CT scan image samples, respectively.

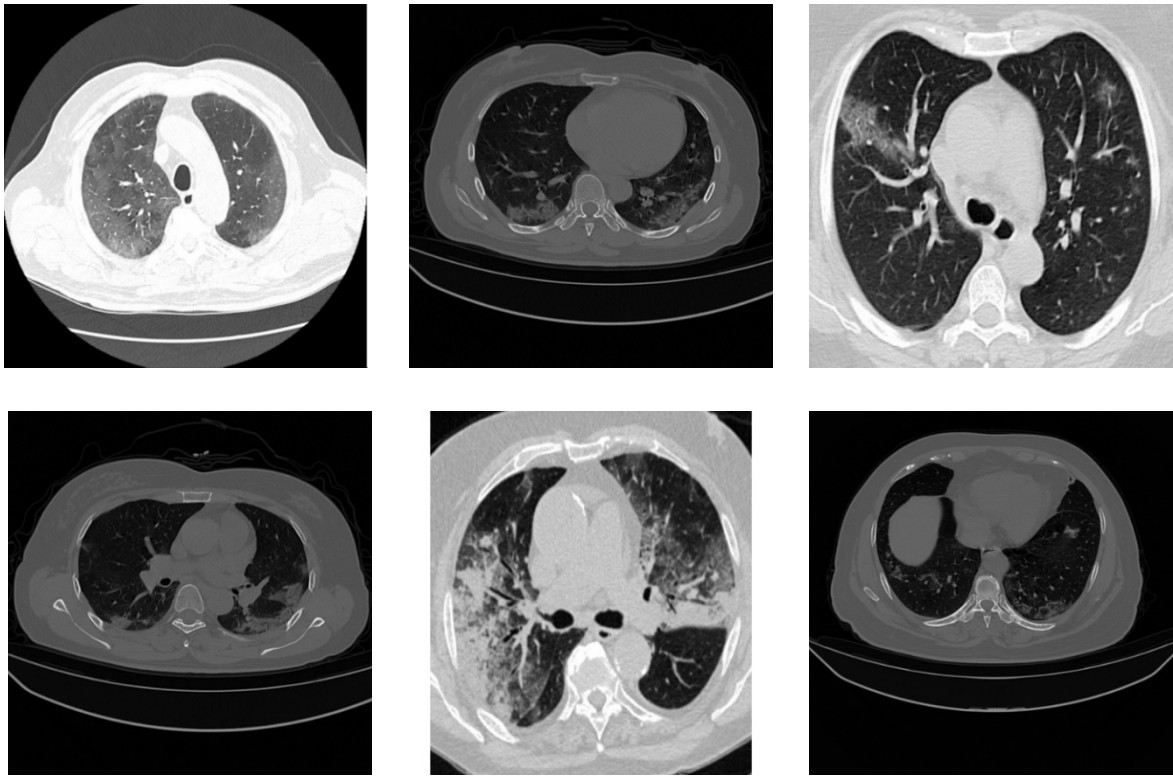

**Figure 2.** COVID 19 Sample CT scans.

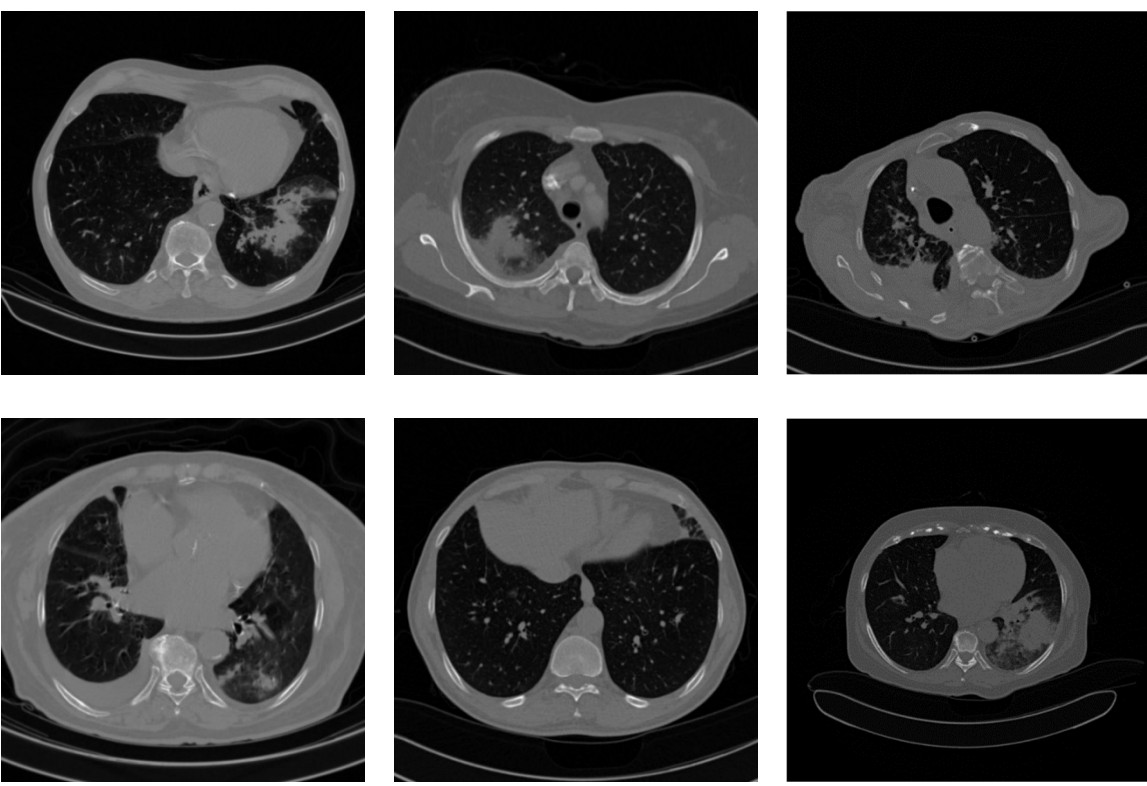

**Figure 3.** Viral Pneumonia Sample CT scans.

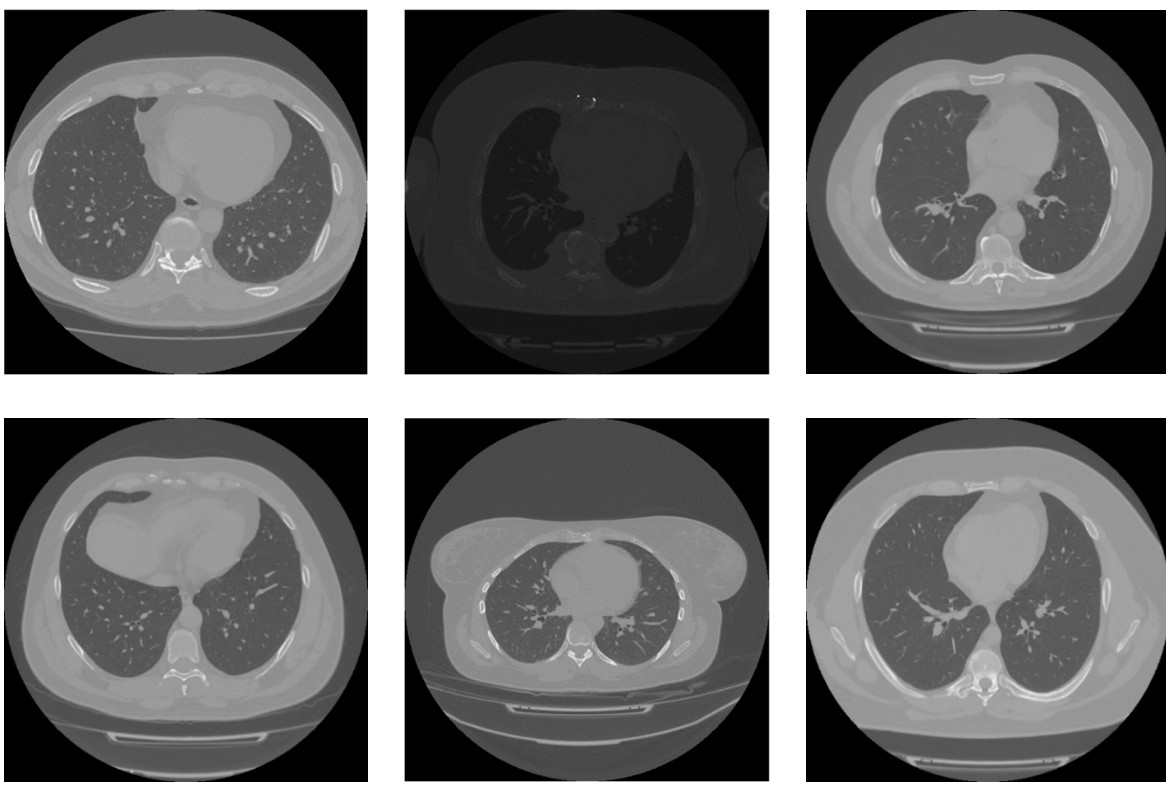

**Figure 4.** Normal Sample CT scans.

## 2.2. Deep CNN Architectures

Convolutional Neural Networks (CNNs) have been traditionally used to extract deep features from image datasets [48] and have been extended to other machine learning tasks



due to their ability to capture complex and positionally invariant feature maps. CNNs train convolutional kernels to extract simple low-level features in shallow layers and complex high-level features in deeper layers. CNNs make use of local receptive fields in which hidden units are connected to local patches of the lower layer. This allows the network to capture local spatial relationships of the image pixels. Weight sharing of the kernels enables translation invariance of the network to objects in images. It also reduces the number of parameters of the network. CNNs also use pooling as a parameter-free down-sampling operation serving to aggregate information. This section briefly discusses different CNN architectures used in this work, such as VGG-19, Inception-V3, ResNet-152-v2, InceptionResNet-V2, DenseNet-201, XceptionNet, and EfficientNet-B7. The Deep CNNs automatically extract important features pertaining to the dataset, thereby removing the need for handcrafted feature engineering.

VGG-19 [39] is a standard deep CNN with 19 convolutional layers. VGG-19 employs stacks of $3 \times 3$ convolutional kernels with max-pooling layers in between them. This produces a simple deep architecture where the $3 \times 3$ kernels extract complex features and the max-pooling layers amplify the important features.

The Inception architecture runs multiple parallel convolutions using various kernels to extract features at various scales [49] as opposed to typical CNNs, in which the convolutions are stacked sequentially. The architecture proposes a wider as well as deeper architecture. The third version of the Inception family is the 48-layer deep Inception-v3, which utilizes $1 \times 1$ pointwise convolution layers to reduce the feature dimensionality and parameters of the model, as well as to improve computational efficiency.

Resnet-152-v2 architecture is 152 layers deep. This version of the architecture performs batch normalization and ReLU activation before the convolution for a better identity mapping [50] and is employed in this paper.

The Inception-ResNet [51] architecture utilizes residual connections between the Inception blocks, allowing for better gradient propagation, thereby improving the Inception network's depth as well as the training speed. Inception-ResNet-v2, which is a 164-layer deep architecture, was utilized in this work.

The DenseNet [52] architecture aims to tackle the vanishing gradient problem in classical CNNs by introducing dense blocks where each convolution output is forward propagated to every other convolution layer through skip connections. This study used the DenseNet-201 architecture, which has 201 layers.

Xception ("Extreme version of Inception") is a 71-layer deep Inception-inspired architecture that utilizes depth wise separable convolutions and $1 \times 1$ pointwise convolutions to substantially decrease the model parameters and improve computation efficiency, thereby preventing the model from overfitting the dataset [53].

EfficientNet proposed a compound factor to consistently scale the baseline model's parameters, such as resolution, width, and depth. This compound scaling allows the model to adapt to different image sizes by introducing additional layers or channels to extract deep features [54]. The baseline model is obtained by performing a Neural Architecture Search on the dataset. EfficientNet-B7 inspired by the architecture searched for ImageNet dataset was utilized in this research.

Ensemble modeling is an approach in which multiple separate models are developed to predict an outcome, either using a variety of modeling techniques or by Bagging the training database. The meta-learners then aggregate each base model's predictions to provide a single final prediction. The purpose of employing Ensemble models is to reduce the generalization error on the test database by reducing the variance.

*2.3. Deep Meta-Leaners*

Ensemble modeling has been used in various fields and has been found to produce enhanced results [55]. Deep learning models used as meta-learners for Ensemble modeling include Neural Network (NN) Architecture, Fully Convolutional Network (FCN) Architecture, and Multi-Head Attention Encoder. Since the former two have been thoroughly

discussed in many articles, a brief overview of Multi-Head Attention Encoder alone is outlined, as the paper proposes a novel approach based on the same.

The Transformer encoder is made of several stacked encoder modules, each of which feeds into the encoder at the next level. Each encoder module employs sublayers of Multi-Head Attention and feedforward networks, where each sublayer adopts a residual connection and layer normalization. The attention encoder output does not depend on the order of inputs. Therefore, positional embeddings were added to the feature embeddings so that information about the order of the sequence is retained. The embedded sequence was passed as query ($Q$), key ($K$), and value ($V$) to the attention encoder module.

Self-Attention allows the model to focus on other embeddings in the sequence that need to be paid attention to while processing one particular embedding. A scaled dot-product attention mechanism was utilized in the self-attention layer, as it obtains more stable gradients by scaling the values to a manageable range.

$$Attention(Q,\ K,\ V) = softmax\left(\frac{Q\ K^T}{\sqrt{d_k}}\right)V \tag{1}$$

Multi-Head Attention concatenates multiple self-attention representations to get the attention output thereby expanding the model's ability to focus on different parts of the sequence. The attention output is then passed on to the next encoder module through a feedforward network with residual connections between sublayers.

Figure 5 illustrates the Multi-Head Attention Encoder architecture.

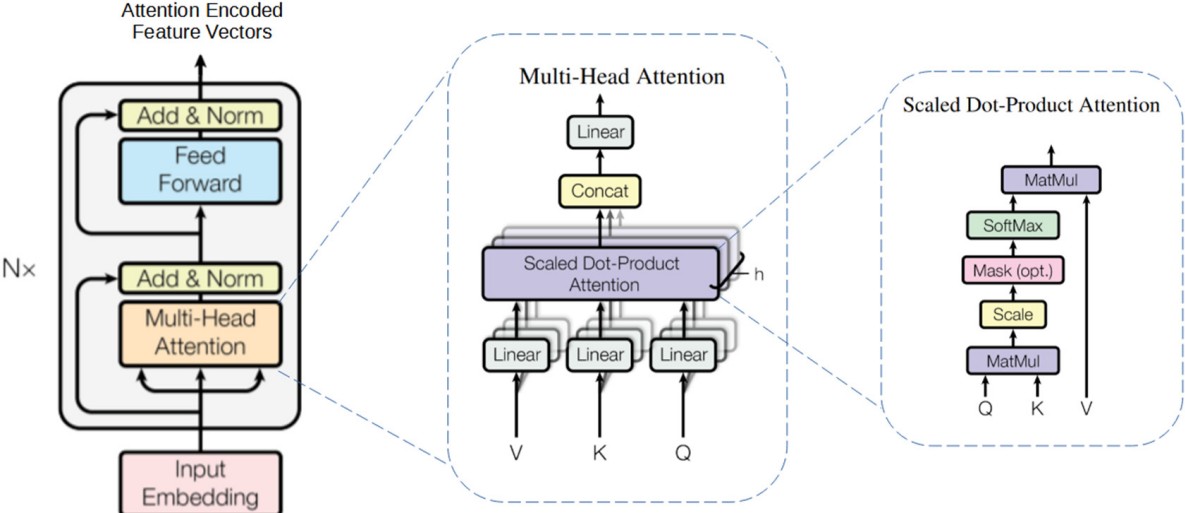

**Figure 5.** Multi-Head Attention Encoder.

### 2.4. Methodology

This section explains the proposed Ensemble Learning methodology illustrated in Figure 6. The proposed method consists of a features extraction layer and a classification layer. Various Deep CNNs are used as feature extractors. The Ensemble models represent chest CT scan classifiers with Deep CNNs as feature-extracting base learners and the presented deep and shallow classifiers as meta-learners. Seven popular Deep CNN architectures were trained and evaluated on the CT scan database. The Deep CNNs were coupled with the densely connected network to classify the chest CT scan images. These models are then trained and hyperparameter tuned to obtain the optimum Deep CNN classifiers. The four top-performing models were used as trained base-feature extractors for Ensemble learning. These Deep CNN feature extractors were used as base learners in the Ensemble. The features were in the form of model probabilities or a deep feature vector or deep feature map. The classification Ensemble models, such as average Ensemble, majority Ensemble, shallow machine learning Ensembles, NN Ensembles, CNN Ensembles

and attention-based Ensembles are used in experiments to synergize the output features from the Deep CNNs and perform the final classification.

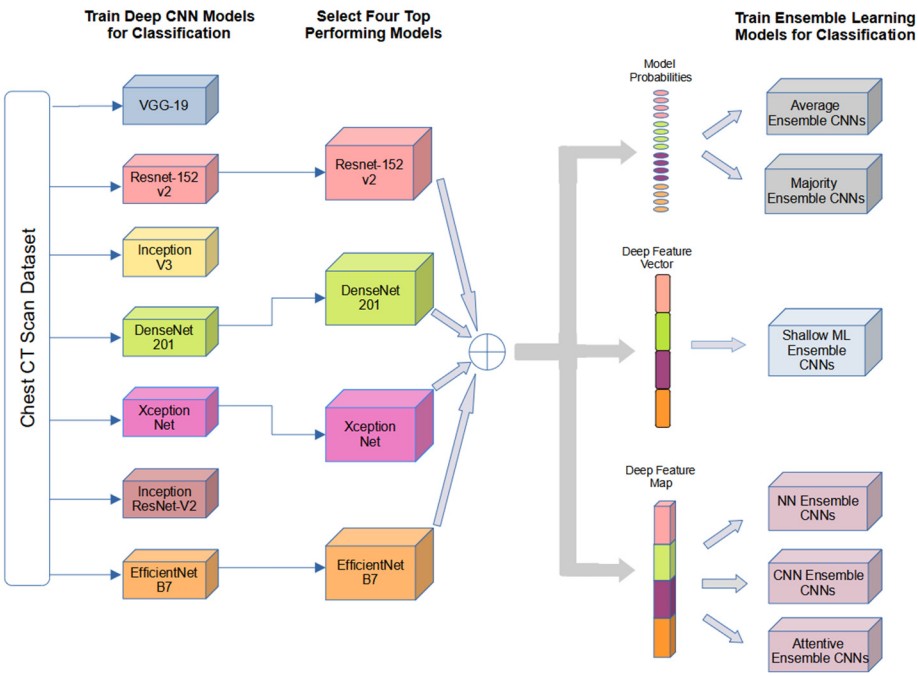

**Figure 6.** Overview of Methodology.

Various state-of-the-art Deep CNN architectures are used to extract deep feature representations from the CT scan image database. VGG-19, Inception-V3 and ResNet-152-v2, Densenet-201, InceptionResNet-V2, Xception, and Efficientnet-B7 are used as feature extractors to obtain deep feature maps that are then used to train a fully connected neural network classification head to obtain the final model prediction. The hyperparameters of the Deep CNN feature extractors are optimized in [39,49,50,52–54]. These architectures are initialized with ImageNet weights for transfer learning on 224 × 224 sized images. The classification head is hyperparameter tuned with different numbers of layers, a different number of neurons per layer and dropout factors. Dropout regularization is used to prevent the classification head from overfitting the training fold database and generalize better on the validation fold. ReLU activation is employed to learn the non-linearity. The CT scan images are first preprocessed by resizing them to 224 × 224 pixels using Bilinear interpolation and rescaling the image intensity to the range [–1, 1] so as to get it into the correct format pertaining to the architectures, thereby making efficient use of transfer learning.

Average and Majority Ensemble CNNs are described in Section 2.4.1. Shallow Ensemble CNNs are described in Section 2.4.2. NN Ensemble CNNs are described in Section 2.4.3. CNN Ensemble CNNs are described in Section 2.4.4. Attentive Ensemble CNNs are described in Section 2.4.5.

A four-layer neural network classification head with 25% dropout regularization gives the best results for VGG-19, Inception-v3, ResNet-152-v2, InceptionResNet-v2 models while a three-layer neural network classification head with 50% dropout regularization gives the best results for Densenet-201, Efficientnet-B7 and Xception models. The models were trained using Adam [55] as an optimizer with a learning rate of 0.001 and the calculated loss being categorical cross-entropy. Figure 7 demonstrates training a Deep CNN model for feature extraction. The top four best-performing architectures obtained on evaluating the test are saved as base models for future Ensemble Deep CNNs models.

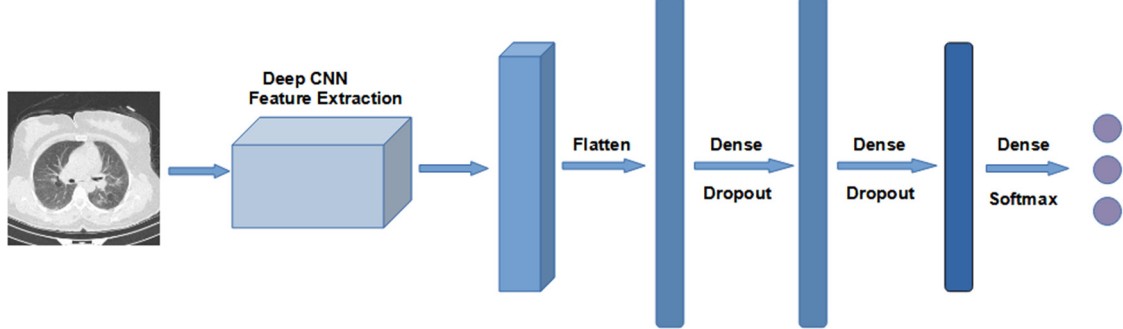

**Figure 7.** Training Deep CNN Models.

2.4.1. Deep Voting Ensemble CNN Models

In the Voting Ensemble CNN Model, as shown in Figure 8, the softmax scores obtained from the four trained classification models corresponding to each class are maxed or averaged to obtain the Ensemble model's predicted class. These models do not require additional training on five-fold cross-validation.

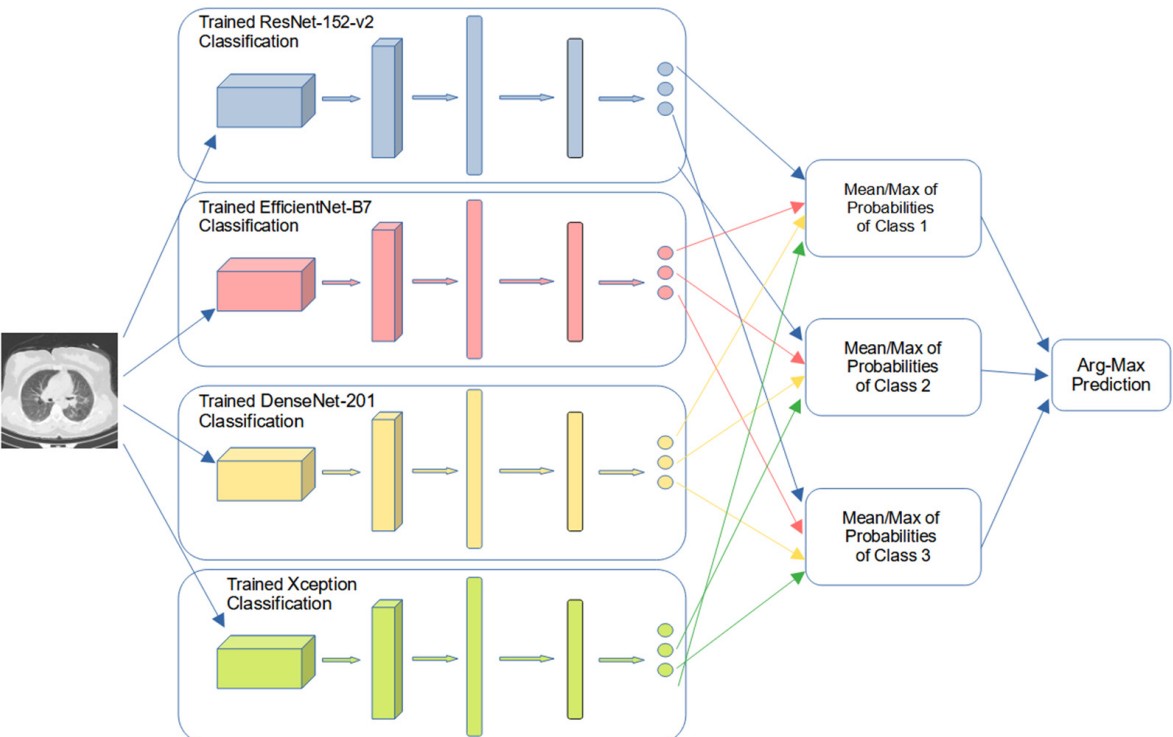

**Figure 8.** Deep Voting Ensemble Model.

2.4.2. Shallow Meta-Learning Ensemble CNN Models

For Deep Ensemble CNNs using Shallow Meta-Learners, the feature vectors obtained from the trained deep CNN feature extractors were first stacked to get a deep feature vector. This deep feature vector was then fed to a machine learning classifier. Logistic Regression, Support Vector Machine, Decision Tree, Gaussian Naive Bayes, K-Nearest-Neighbor (KNN), Random Forest Ensemble, Bagging Ensemble, AdaBoost Ensemble, and Gradient Boost Ensemble models were leveraged as meta-learning classifiers for the deep ensemble approach. The Logistic Ensemble CNNs Model was trained with a regularizing strength of 40 and l2 penalization. The model was trained for 100 iterations for convergence. The SVC Ensemble CNNs Model gave the best results using the RBF kernel, regularizing strength of 40 and l2 penalization. The Decision Tree Ensemble CNNs Model was hyper

parameterized using the Gini impurity criterion for splits and the tree was allowed to deepen till all leaves were pure or had two samples. The Naïve Bayes Ensemble CNNs Model used the Gaussian Naïve Bayes algorithm for classification. The KNN Ensemble CNNs Model was trained with K = 10 obtained using the elbow method. The Random Forest Ensemble CNNs Model was tuned to 100 tree estimators without bootstrapping dataset and Gini impurity as the split condition to get optimum results. Bagging Ensemble CNNs Model using 30 Decision tree base estimators and a bootstrapped dataset gave the best result on the test database. The AdaBoost Ensemble Model with 100 Decision Tree-based learners and a learning rate of 0.7 gave optimum results. The Gradient Boosting Ensemble Model was hyper parameterized with 100 base estimator boosting rounds with a learning rate of 0.9 for the best results. Figure 9 depicts the proposed approach.

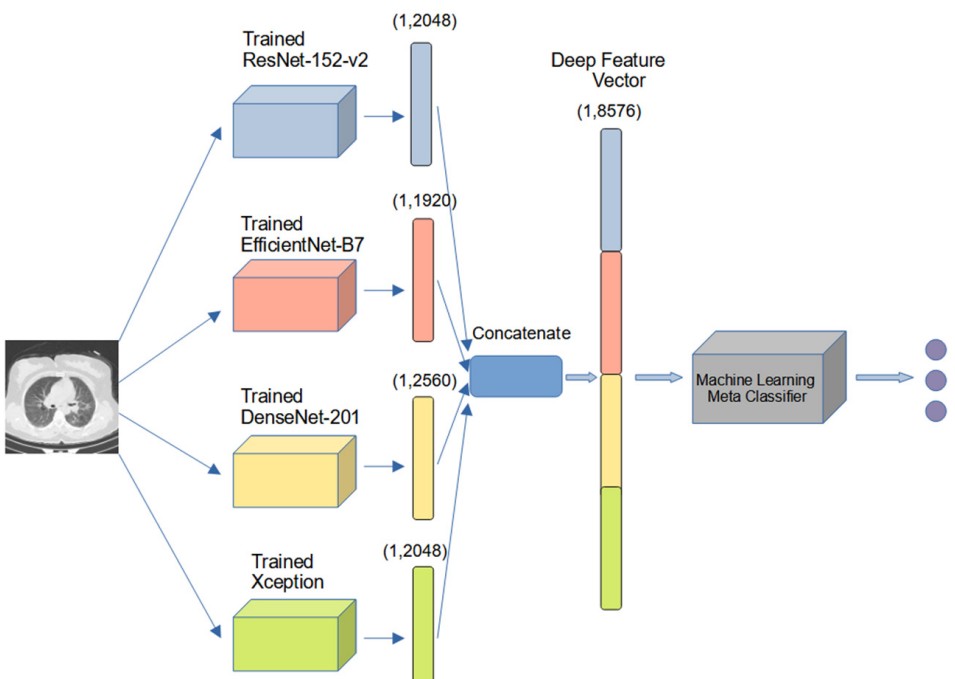

**Figure 9.** Shallow Meta-Learning Ensemble CNN Model.

### 2.4.3. NN Ensemble CNN Model

In the NN Ensemble CNN meta-learning approach [56], the feature maps obtained from the final convolutional layer of the four trained Deep CNN models were global average pooled to get deep feature vectors. Global Average Pooling takes the average of each feature map, thereby reducing the size of the features and the parameters to get a better representational mapping to the class labels. These four feature vectors were then combined to get the Ensemble deep feature vectors.

A feed-forward neural network (NN) classification head was then employed to meta-learn the deep ensemble feature vector along with the final softmax activation, which gives the class probabilities. The NN classification head was hyperparameter tuned with different numbers of layers, number of neurons per layer, and dropout factors. After tuning, the best validation accuracy was obtained using a four-layer NN with 1024, 256, 64 and 3 neurons, respectively, to each layer and 0.2% dropout, which is shown in Figure 10. The maximum of the softmax scores was used as the Ensemble model's predicted class for classification.

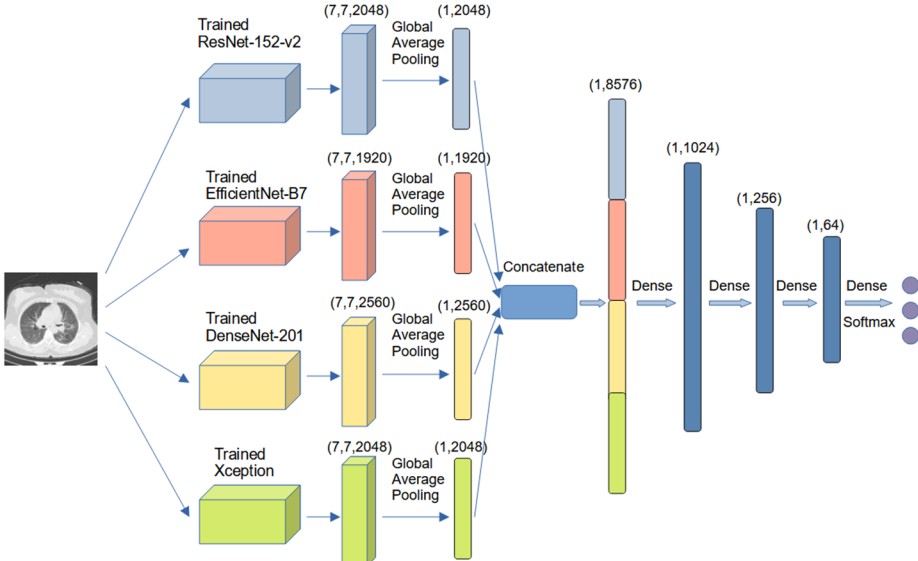

**Figure 10.** NN Ensemble CNN Model.

### 2.4.4. FCN Ensemble CNN Model

In the FCN Ensemble CNN meta-learning approach [57], feature map outputs from the final convolutional layer of the four trained Deep CNN models were then concatenated along the channels to obtain the Ensemble deep feature map. A Fully Convolutional Network (FCN) was then meta-learned to obtain the final softmax predictions. The FCN architecture was hyperparameter tuned with different convolutional layers and different numbers of various convolutional and max-pooling kernel sizes ($2 \times 2$, $3 \times 3$ and $4 \times 4$). After experimentation, an FCN modeled with three convolutional blacks made of $2 \times 2$ convolutional and max-pooling kernels of 2048, 512, and 64 kernels per block, respectively gave the best result on the validation database. Figure 11 illustrates the proposed approach. The final feature map was then flattened and passed through a dense layer with softmax activation to get the final classification scores. The maximum of the softmax scores was used as the Ensemble model's predicted class for classification.

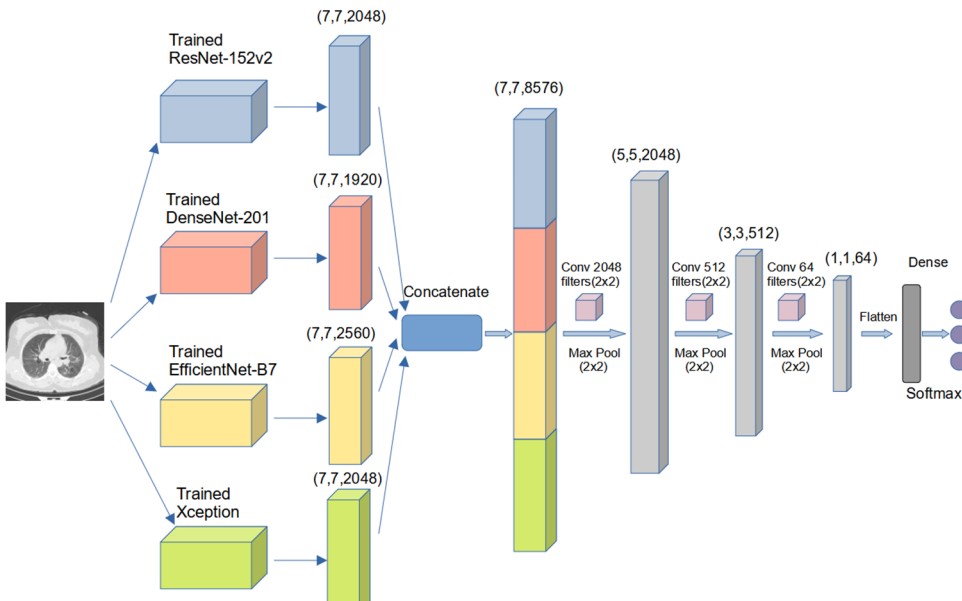

**Figure 11.** FCN Ensemble CNN Model.

### 2.4.5. Attentive Ensemble CNN Model

In the Attentive Ensemble-CNN model, the feature maps corresponding to the final convolutional layers were extracted from the four trained models. First, $1 \times 1$ pointwise convolution was performed on each of these feature maps to reduce the channels of the maps to the same depth. These maps were then passed as input feature embedding to three stacks of Multi-Head Attention encoders. The encoder was constructed using a multi-head attention layer that used a scaled dot product attention mechanism and a Feed Forward Network made of a $3 \times 3$ convolution layer. Skip connections were also employed between the sublayers for better feature propagation. Each of these encoders learned the relative importance of the feature embeddings with respect to the other embeddings. The attention-encoded feature maps were then passed onto the next encoder in a feed-forward fashion as shown in Figure 12. The final encoded feature maps were then concatenated and passed through a dense layer with softmax activation to get the final classification scores. The maximum of the softmax scores assigned the Ensemble model's predicted class for classification.

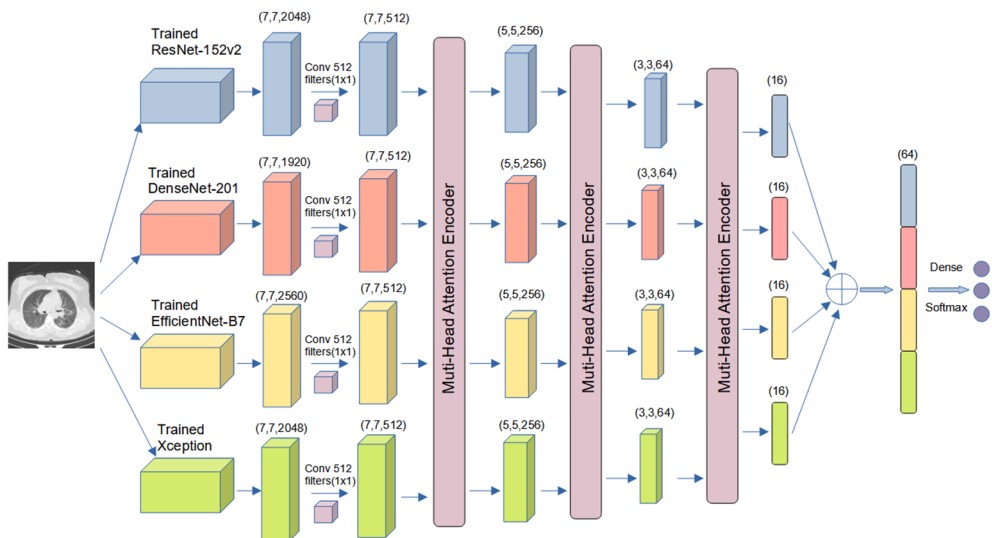

**Figure 12.** Attentive Ensemble CNN Model.

### 2.5. Experiment Setup

The Deep CNN models and the Ensemble learning models were built, trained, and evaluated using the TensorFlow/Keras framework with Python programming on a cloud TPU engine (eight TPU v2 cores with 64 GB total memory performing 180 TFlops). The image database was split into five-fold cross validation and test databases, as shown in Table 1, using a pseudo random generator and loaded into the TPU cache memory. We used 95% of the database for five-fold cross validation. The remaining 5% of the database was used for testing. In the five-fold cross validation dataset, 80% was used as training and 20% was used for cross validation during each fold. Each model was trained using five-fold cross validation before being restored to the epoch with the best results on the validation database. To exploit TPU performance, images with a resolution of $224 \times 224$ pixels were sent to the models in mini batches of 80.

**Table 1.** Five-fold cross validation and testing CT scan image distribution.

|  | CT Scan Database Split | Count |
| --- | --- | --- |
|  | Training | 13,000 |
| 5-Fold Cross Validation | Validation | 3248 |
|  | Testing | 856 |

## 3. Results

### 3.1. Evaluating Single Deep CNN Architectures

An Adam optimizer [56] with a learning rate of 0.001 was utilized to train the Deep CNN classification models using categorical cross-entropy as the computed loss. The evaluation metrics obtained by these models on the test database are shown in Table 2. It is evident from the table that ResNet-152-v2, Densenet-201, Efficientnet-B7, and Xception architectures gave better accuracy, average F1-score, and average MCC on the test database compared to VGG-19, Inception-v3, and InceptionResNet-v2. The four best-performing models were saved and used as base models for future Ensemble learning approaches. The Xception Model gave the best accuracy of 99.18%, while the VGG-19 model gave the worst accuracy of 96.18%. All the models except VGG-19 were able to classify the CT scan images with accuracy and F1-score more than 98.65%. The Inception-v3, ResNet-152-v2, Efficientnet-B7, and Xception models were able to classify the Viral Pneumonia CT scan images with a perfect score of one. The DenseNet-201, ResNet-152-v2, Xception, and EfficientNet-b7 models performed fairly even at high thresholds, producing an AUC value greater than 99.2%.

**Table 2.** Evaluation Metrics for Deep CNN Models.

|  | Accuracy | Class Label | Precision | Recall | F1 Score | AUC | MCC |
|---|---|---|---|---|---|---|---|
| VGG-19Model | 0.9678 | COVID-19 | 0.9810 | 0.9453 | 0.9628 | 0.9585 | 0.9607 |
|  |  | Normal | 0.9430 | 0.9821 | 0.9622 | 0.9525 | 0.9597 |
|  |  | Viral Pneumonia | 0.9926 | 0.9926 | 0.9926 | 0.9956 | 0.9910 |
| Inception-v3Model | 0.9871 | COVID-19 | 0.9946 | 0.9765 | 0.9855 | 0.9879 | 0.9844 |
|  |  | Normal | 0.9738 | 0.9940 | 0.9838 | 0.9849 | 0.9827 |
|  |  | Viral Pneumonia | 1.0000 | 1.0000 | 1.0000 | 1.0000 | 1.0000 |
| ResNet-152-v2Model | 0.9895 | COVID-19 | 0.9921 | 0.9843 | 0.9882 | 0.9897 | 0.9863 |
|  |  | Normal | 0.9823 | 0.9910 | 0.9867 | 0.9882 | 0.9848 |
|  |  | Viral Pneumonia | 1.0000 | 1.0000 | 1.0000 | 1.0000 | 1.0000 |
| Inception ResNet-v2Model | 0.9883 | COVID-19 | 1.0000 | 0.9765 | 0.9881 | 0.9906 | 0.9871 |
|  |  | Normal | 0.9795 | 0.9970 | 0.9882 | 0.9888 | 0.9875 |
|  |  | Viral Pneumonia | 0.9782 | 1.0000 | 0.9890 | 0.9891 | 0.9869 |
| DenseNet-201Model | 0.9884 | COVID-19 | 0.9794 | 0.9947 | 0.9870 | 0.9875 | 0.9863 |
|  |  | Normal | 0.9945 | 0.9792 | 0.9871 | 0.9903 | 0.9865 |
|  |  | Viral Pneumonia | 1.0000 | 0.9925 | 0.9962 | 0.9993 | 0.9959 |
| Xception Model | 0.9918 | COVID-19 | 0.9947 | 0.9870 | 0.9908 | 0.9921 | 0.9895 |
|  |  | Normal | 0.9852 | 0.9940 | 0.9896 | 0.9907 | 0.9873 |
|  |  | Viral Pneumonia | 1.0000 | 1.0000 | 1.0000 | 1.0000 | 1.0000 |
| EfficientNet-b7Model | 0.9895 | COVID-19 | 0.9947 | 0.9817 | 0.9867 | 0.9900 | 0.9843 |
|  |  | Normal | 0.9795 | 0.9940 | 0.9867 | 0.9878 | 0.9846 |
|  |  | Viral Pneumonia | 1.0000 | 1.0000 | 1.0000 | 1.0000 | 1.0000 |

### 3.2. Evaluating Voting Ensemble CNN Models

For the Average and Majority Voting Ensemble models, the four saved trained models (ResNet-152-v2, Densenet-201, Xception and Efficientnet-b7) were used as base models. The Voting Ensemble model, evaluated on the test database, produced the evaluation metrics as shown in Table 3. Both the models produced an accuracy and average F1 score greater than 99.4%, thereby outperforming the deep CNN architectures. Both the models performed relatively well even at high thresholds by producing an AUC value greater than 99.56%, outperforming the Deep CNN models.

**Table 3.** Evaluation Metrics for Voting Ensemble CNN Models.

| | Accuracy | Class Label | Precision | Recall | F1 Score | AUC | MCC |
|---|---|---|---|---|---|---|---|
| Average Ensemble CNN Model | 0.9953 | COVID-19 | 1.0000 | 0.9895 | 0.9947 | 0.9957 | 0.9931 |
| | | Normal | 0.9882 | 1.0000 | 0.9941 | 0.9941 | 0.9928 |
| | | Viral Pneumonia | 1.0000 | 1.0000 | 1.0000 | 1.0000 | 1.0000 |
| Majority Ensemble CNN Model | 0.9942 | COVID-19 | 1.0000 | 0.9869 | 0.9934 | 0.9947 | 0.9932 |
| | | Normal | 0.9853 | 1.0000 | 0.9926 | 0.9926 | 0.9921 |
| | | Viral Pneumonia | 1.0000 | 1.0000 | 1.0000 | 1.0000 | 1.0000 |

### 3.3. Evaluating Shallow Meta-Learning Ensemble CNN Models

For the Shallow meta-learning Ensemble CNNs models, the four trained Deep CNN feature extractors were used as Ensemble base models to train the machine learning meta-learners, as shown in Figure 5. Table 4 shows the metrics obtained after evaluating the models on the test database. The Logistic, KNN, Random Forest, Bagging, AdaBoost and Gradient Boosting meta learners achieved accuracy and an average F1 score of more than 99.18%, thereby outperforming the deep CNN architectures. Random Forest and Gradient Boosting Ensemble CNNs models produced the best performance but were not able to outperform the Voting Ensemble CNNs models. The KNN, Random Forest, AdaBoost, Gradient Boosting, and Bagging Ensemble CNNs models performed relatively well even at high thresholds by producing an AUC value greater than 99.5%, outperforming the Deep CNN models. The rest of the shallow meta-learners performed relatively poorly at higher classification thresholds. The Shallow meta-learning Ensemble CNNs models were not able to outperform the Voting Ensemble CNNs models in terms of AUC.

**Table 4.** Evaluation Metrics for Deep Ensemble Models using ML classifiers.

| | Accuracy | Class Label | Precision | Recall | F1 Score | AUC | MCC |
|---|---|---|---|---|---|---|---|
| Logistic Ensemble CNNs Model | 0.9918 | COVID-19 | 0.9896 | 0.9921 | 0.9908 | 0.9916 | 0.9889 |
| | | Normal | 0.9910 | 0.9881 | 0.9895 | 0.9916 | 0.9890 |
| | | Viral Pneumonia | 1.0000 | 1.0000 | 1.0000 | 1.0000 | 1.0000 |
| SVC Ensemble CNNs Model | 0.9885 | COVID-19 | 0.9795 | 0.9973 | 0.9883 | 0.9874 | 0.9875 |
| | | Normal | 0.9969 | 0.9762 | 0.9865 | 0.9899 | 0.9851 |
| | | Viral Pneumonia | 1.0000 | 1.0000 | 1.0000 | 1.0000 | 1.0000 |
| Decision Tree Ensemble CNNs Model | 0.9907 | COVID-19 | 0.9947 | 0.9843 | 0.9895 | 0.9910 | 0.9892 |
| | | Normal | 0.9824 | 0.9940 | 0.9882 | 0.9892 | 0.9879 |
| | | Viral Pneumonia | 1.0000 | 1.0000 | 1.0000 | 1.0000 | 1.0000 |
| Naïve Bayes Ensemble CNNs Model | 0.9883 | COVID-19 | 0.9794 | 0.9947 | 0.9870 | 0.9875 | 0.9863 |
| | | Normal | 0.9939 | 0.9792 | 0.9865 | 0.9903 | 0.9859 |
| | | Viral Pneumonia | 1.0000 | 0.9925 | 0.9962 | 0.9993 | 0.9961 |
| KNN Ensemble CNNs Model | 0.9929 | COVID-19 | 0.9922 | 0.9947 | 0.9934 | 0.9939 | 0.9913 |
| | | Normal | 0.9940 | 0.9881 | 0.9910 | 0.9931 | 0.9902 |
| | | Viral Pneumonia | 0.9926 | 1.0000 | 0.9963 | 0.9963 | 0.9947 |
| Random Forest Ensemble CNNs Model | 0.9953 | COVID-19 | 0.9911 | 0.9970 | 0.9940 | 0.9955 | 0.9926 |
| | | Normal | 0.9973 | 0.9921 | 0.9947 | 0.994 | 0.9930 |
| | | Viral Pneumonia | 1.0000 | 1.0000 | 1.0000 | 1.0000 | 1.0000 |
| Bagging Ensemble CNNs Model | 0.9919 | COVID-19 | 0.9947 | 0.9869 | 0.9908 | 0.9921 | 0.9898 |
| | | Normal | 0.9852 | 0.9940 | 0.9896 | 0.9907 | 0.9881 |
| | | Viral Pneumonia | 1.0000 | 1.0000 | 1.0000 | 1.0000 | 1.0000 |
| AdaBoost Ensemble CNNs Model | 0.9941 | COVID-19 | 0.9973 | 0.9895 | 0.9934 | 0.9944 | 0.9925 |
| | | Normal | 0.9882 | 0.9970 | 0.9926 | 0.9931 | 0.9907 |
| | | Viral Pneumonia | 1.0000 | 1.0000 | 1.0000 | 1.0000 | 1.0000 |
| Gradient Boosting Ensemble CNNs Model | 0.9952 | COVID-19 | 1.0000 | 0.9895 | 0.9947 | 0.9941 | 0.9943 |
| | | Normal | 0.9882 | 1.0000 | 0.9941 | 0.9957 | 0.9938 |
| | | Viral Pneumonia | 1.0000 | 1.0000 | 1.0000 | 1.0000 | 1.0000 |

### 3.4. Evaluating Deep Meta-Learning Ensemble CNN Models

For the proposed Deep meta-learning Ensemble CNNs models, the four trained Deep CNN feature extractors were used as ensemble base models to train the deep learning architectures using hyperparameters, as shown in Figures 6–8. An Adam optimizer [56] with a learning rate of 0.001 was utilized to train the ensemble classification models using categorical cross-entropy as the computed loss. Table 5 shows the evaluation metrics obtained after evaluating the models on the test database. The deep meta learners produced an accuracy and average F1 score greater than 99.65%, thereby outperforming the Deep CNN models, Voting Ensemble CNNs models, and Shallow meta-learning Ensemble CNNs models. The Attentive Ensemble model achieved the best performance with an accuracy of 99.88%, classifying COVID-19 with a F1-score of 99.87.

**Table 5.** Evaluation Metrics for Deep Ensemble Models using deep learning classifiers.

|  | Accuracy | Class Label | Precision | Recall | F1 Score | AUC | MCC |
|---|---|---|---|---|---|---|---|
| NN Ensemble CNNs Model | 0.9965 | COVID-19 | 0.9948 | 0.9973 | 0.9960 | 0.9963 | 0.9951 |
|  |  | Normal | 0.9970 | 0.9940 | 0.9955 | 0.9965 | 0.9940 |
|  |  | Viral Pneumonia | 1.0000 | 1.0000 | 1.0000 | 1.0000 | 1.0000 |
| FCN Ensemble CNNs Model | 0.9977 | COVID-19 | 1.0000 | 0.9948 | 0.9973 | 0.9978 | 0.9959 |
|  |  | Normal | 0.9941 | 1.0000 | 0.9970 | 0.9970 | 0.9961 |
|  |  | Viral Pneumonia | 1.0000 | 1.0000 | 1.0000 | 1.0000 | 1.0000 |
| Attentive Ensemble CNNs Model | 0.9988 | COVID-19 | 0.9974 | 1.0000 | 0.9987 | 0.9987 | 0.9978 |
|  |  | Normal | 1.0000 | 0.9970 | 0.9985 | 0.9990 | 0.9980 |
|  |  | Viral Pneumonia | 1.0000 | 1.0000 | 1.0000 | 1.0000 | 1.0000 |

### 3.5. Confusion Matrix for Deep CNN Models

Figure 13a–g depicts the confusion matrices obtained on evaluating the Deep CNN models on the test database. The DenseNet-201 and Xception models performed fairly when classifying the COVID-19 lung CT scans, misclassifying less than five. The Inception-v3, InceptionResNet-v2, Xception, and EfficientNet-b7 models performed fairly when classifying the normal lung CT scans, misclassifying less than three.

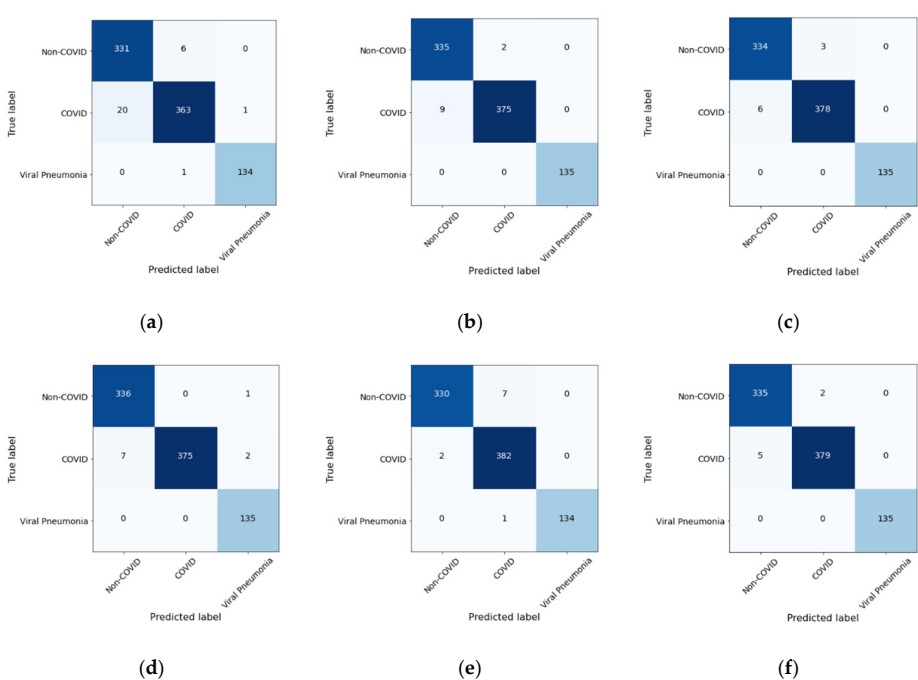

**Figure 13.** *Cont.*

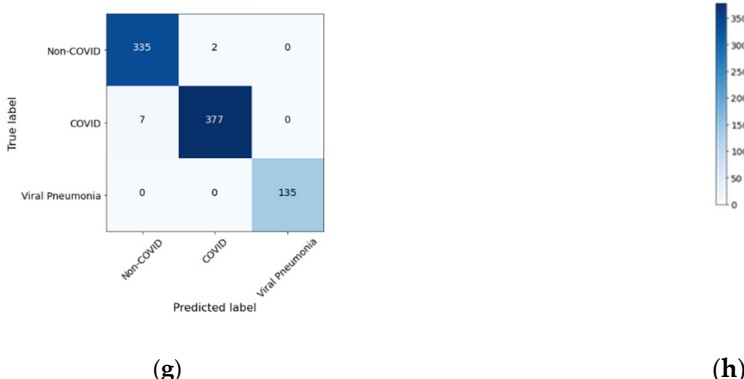

(**g**)                                                                 (**h**)

**Figure 13.** Confusion Matrix for Deep CNN Models (**a**) VGG-19 Model, (**b**) Inception-v3 Model, (**c**) ResNet-152-v2 Model, (**d**) InceptionResNet-v2 Model, (**e**) DenseNet-201 Model, (**f**) Xception Model, (**g**) EfficientNet-b7 Model, (**h**) label.

### 3.6. Confusion Matrix for Voting Ensemble Models

Figure 14a,b show the confusion matrices obtained by evaluating the Voting Ensemble CNNs models on the test database. Both models performed well by perfectly classifying the normal and viral pneumonia lung CT scans and misclassifying less than six COVID-19 lung CT scans.

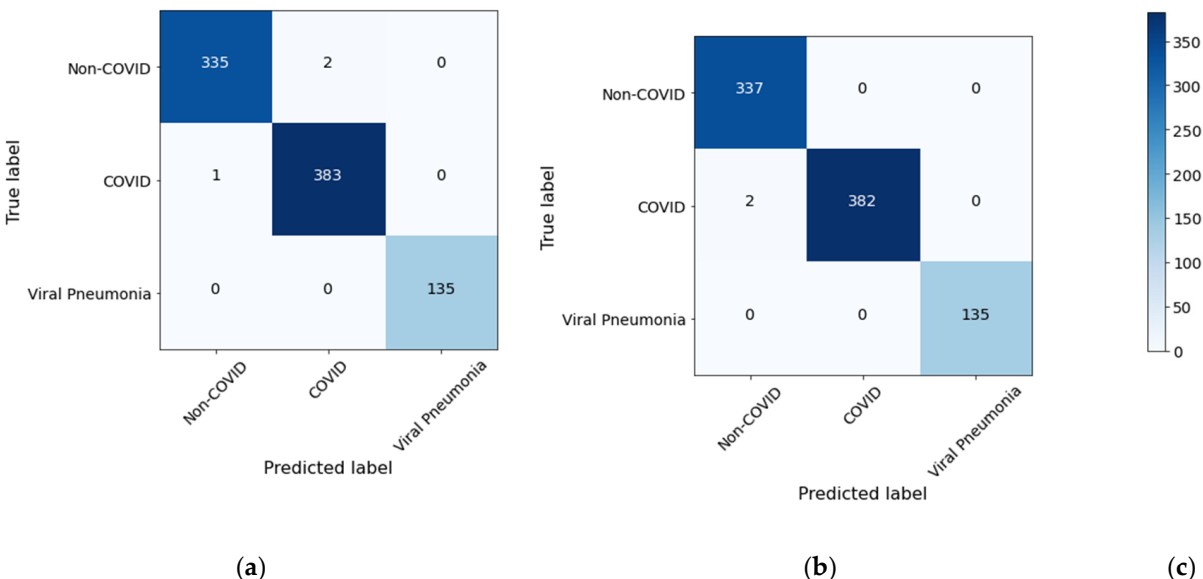

(**a**)                                                     (**b**)                                          (**c**)

**Figure 14.** Confusion Matrix for (**a**) Average Ensemble CNN Model, (**b**) Majority Ensemble CNN Model. (**c**) Label.

### 3.7. Confusion Matrix for Shallow Meta-Learning Ensemble CNN Models

Figure 15a–i show the confusion matrices obtained when evaluating Shallow meta-learning Ensemble CNNs models on the test database. The SVC, KNN, Naive Bayes, and Random Forest Ensemble CNNs models performed really well when classifying the COVID-19 lung CT scans, misclassifying less than four. The Decision Tree, Random Forest, AdaBoost, Gradient Boosting, and Bagging Ensemble CNNs models performed well when classifying the normal lung CT scans, misclassifying less than three. All these models misclassified one or zero viral pneumonia lung CT scans.

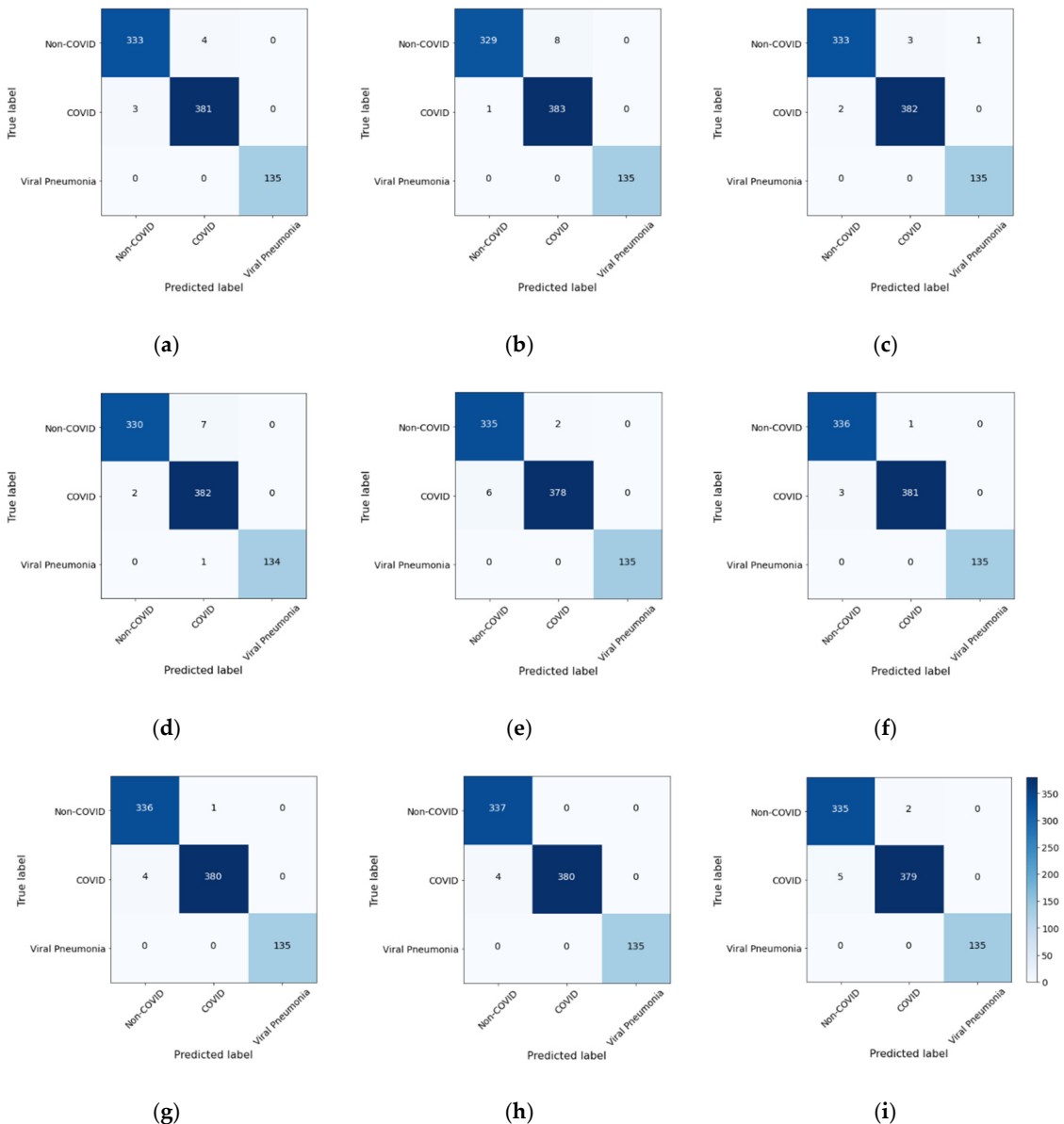

**Figure 15.** Confusion Matrix for Shallow Meta-Learning Ensemble CNN Models (**a**) Logistic, (**b**) SVC, (**c**) KNN, (**d**) Naïve Bayes, (**e**) Decision Tree, (**f**) Random Forest, (**g**) AdaBoost, (**h**) Gradient Boosting, (**i**) Bagging Ensemble CNN Model.

*3.8. Confusion Matrix for Deep Meta-Learning Ensemble CNN Models*

Figure 16a–c shows the confusion matrices obtained by evaluating the Deep meta-learning Ensemble CNNs models on the test database. From the confusion matrices, it is evident that these models achieved the best performance, misclassifying less than four. The Attentive Ensemble CNNs model perfectly classified all the COVID-19 lung CT scan images, while the FCN Ensemble CNNs model perfectly classified all the normal lung CT scan images. All the models perfectly classified all the viral pneumonia test lung CT scans. The models produced the best results even at high thresholds by achieving an AUC value greater than 99.75%, outperforming the Deep CNN models and Shallow meta-learning Ensemble CNNs models.

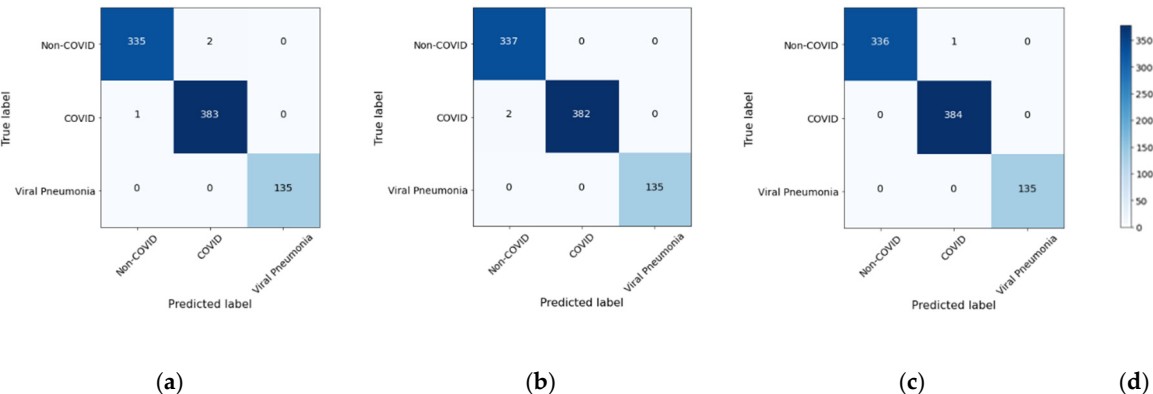

(**a**)         (**b**)         (**c**)    (**d**)

**Figure 16.** Confusion Matrix for Deep Meta-Learning Ensemble CNN Models (**a**) NN, (**b**) FCN, (**c**) Attentive Ensemble CNN Model, (**d**) label.

*3.9. Robustness on Chinese CT Scan Database*

The robustness of the proposed framework to generalization was tested by evaluating the trained model on a subset of a Chinese CT Scan database [58]. First, 1470 CT scan images were tested and classified as COVID infection, common pneumonia, or Normal condition. Table 6 shows the accuracy and F1 score of the evaluated models on the Chinese CT Scan database. The Deep CNN models struggled to classify some images due to the variance in the Chinese data distribution compared to the training database. The proposed models were able to perform decently, thereby ensuring the generalization capability of the Ensemble approach.

**Table 6.** Evaluation Metrics on Chinese CT Scan database.

| Model | Accuracy | F1 Score | MCC |
|---|---|---|---|
| VGG-19 | 0.8880 | 0.8879 | 0.8872 |
| Inception-v3 | 0.8977 | 0.8974 | 0.8968 |
| ResNet-152-v2 | 0.9253 | 0.9251 | 0.9250 |
| InceptionResNet-v2 | 0.9499 | 0.9498 | 0.9494 |
| DenseNet-201 | 0.9482 | 0.9481 | 0.9480 |
| Xception | 0.9511 | 0.9508 | 0.9499 |
| EfficientNet-b7 | 0.8794 | 0.8794 | 0.8790 |
| Average Ensemble CNNs | 0.9641 | 0.9641 | 0.9639 |
| Majority Ensemble CNNs | 0.9710 | 0.9698 | 0.9698 |
| Logistic Ensemble CNNs | 0.9402 | 0.9401 | 0.9438 |
| SVC Ensemble CNNs | 0.9395 | 0.9394 | 0.9394 |
| Decision Tree Ensemble CNNs | 0.9513 | 0.9511 | 0.9498 |
| Naïve Bayes Ensemble CNNs | 0.9381 | 0.9369 | 0.9364 |
| KNN Ensemble CNNs | 0.9290 | 0.9287 | 0.9286 |
| Random Forest Ensemble CNNs | 0.9644 | 0.9643 | 0.9643 |
| Bagging Ensemble CNNs | 0.9597 | 0.9597 | 0.9597 |
| AdaBoost Ensemble CNNs | 0.9644 | 0.9643 | 0.9642 |
| Gradient Boosting Ensemble CNNs | 0.9750 | 0.9747 | 0.9744 |
| NN Ensemble CNNs | 0.9766 | 0.9765 | 0.9760 |
| FCN Ensemble CNNs | 0.9749 | 0.9749 | 0.9748 |
| Attentive Ensemble CNNs | 0.9784 | 0.9782 | 0.9778 |

*3.10. Model Complexities*

Table 7 shows the total parameters and the inference time (at 180 TFlops) of each of the presented Deep Learning models. The time complexity of the proposed approach includes time taken to train four base CNNs, as well as the meta-learners. The space complexity of the frameworks includes storing the CT scan images as input for Deep CNNs, and also the output deep feature maps, which will then serve as input for the meta-learners. The trained parameters of the base-learners and the meta-learners need to

be saved for inference. The hyperparameters of the Deep CNN models are as proposed in works [48–53]. The Voting Ensemble CNNs models used the parameters of the entire Deep CNN classification models for final prediction, while the presented Deep and Shallow Meta-Learning Ensemble CNNs models utilized the parameters of the Deep CNN feature extractors for meta-learning. The inference time was obtained after warning the TPU for a few epochs on the test database. The Voting Ensemble CNNs Models used the least inference time while the Deep meta-learning Ensemble CNNs models consumed the most inference time.

**Table 7.** Model Complexities.

| Model | Abbreviation | Model Type | Total Parameters | Inference Time (ms/image) |
|---|---|---|---|---|
| VGG-19 | VGG | | 46,273,347 | 10.531 |
| Inception-v3 | IV3 | | 74,790,435 | 6.062 |
| ResNet-152-v2 | R152 | | 161,650,947 | 6.710 |
| InceptionResNet-v2 | IRV2 | CNN | 94,217,187 | 9.500 |
| DenseNet-201 | D201 | | 115,218,755 | 11.34 |
| Xception | XCEP | | 124,180,779 | 7.093 |
| EfficientNet-b7 | ENB7 | | 193,107,098 | 10.718 |
| Average Ensemble CNNs | AEC | Voting | 413,133,967 | 4.211 |
| Majority Ensemble CNNs | MEC | Ensemble | 413,133,967 | 4.208 |
| Logistic Ensemble CNNs | LEC | | 161,627,799 | 4.615 |
| SVC Ensemble CNNs | SVC | | 161,628,476 | 4.887 |
| Decision Tree Ensemble CNNs | DTE | | 161,620,824 | 7.231 |
| Naïve Bayes Ensemble CNNs | NBE | Shallow | 161,620,015 | 4.712 |
| KNN Ensemble CNNs | KNN | Ensemble | 161,620,324 | 4.698 |
| Random Forest Ensemble CNNs | RFEC | | 161,621,950 | 7.723 |
| Bagging Ensemble CNNs | BEC | | 161,624,916 | 8.072 |
| AdaBoost Ensemble CNNs | ADA | | 161,624,377 | 8.588 |
| Gradient Boosting Ensemble CNNs | GBEC | | 161,625,412 | 8.974 |
| NN Ensemble CNNs | NNEC | Deep | 170,708,994 | 13.215 |
| FCN Ensemble CNNs | FCNEC | Ensemble | 236,229,890 | 14.327 |
| Attentive Ensemble CNNs | ATEC | | 174,006,978 | 14.671 |

## 4. Discussion

Ensemble modeling combined the four trained deep CNNs, allowing the meta-learners to decide predictions based on different features learned using the base models. The deep features were extracted from four trained models, thereby ensuring independence between the features during Ensemble modeling. Comparing Tables 3–5 with Table 2, it is evident that the presented Deep Ensemble CNNs models outperformed the deep CNN architectures, portraying better accuracy, F1-scores and MCC scores. In contrast with the Conventional Ensemble Learning Models that aggregate the final predictions from the base models to produce the ensemble prediction, the Deep Ensemble Learning Models use deep feature vectors or feature maps from the trained models to train shallow or deep meta-learners. From the experiments, it can be deduced that using an Ensemble model with a strong meta-classifier such as NN, CNN, or Attention can outperform the Conventional and Shallow Ensemble models. The improved model complexity also allows the model to fit better on the training data, thereby reducing the bias. By synergizing the deep feature maps, the Deep Ensemble models learn broad independent features pertaining to the data, thereby allowing the models to generalize better on the test database.

Figures 17 and 18 summarizes the prediction accuracies and COVID-19 classification F1-scores obtained after evaluating the models on the test database. Out of the ensemble approaches, the proposed Attentive Ensemble CNNs Model produced better generalization in different evaluation metrics, as it allowed the model to learn the relative significance between the ensemble feature embeddings.

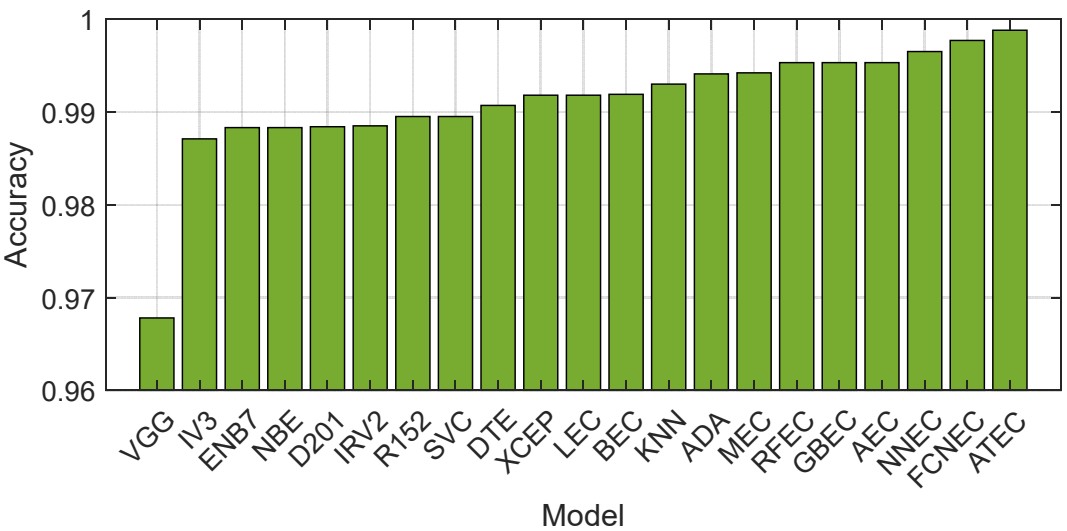

**Figure 17.** Prediction Accuracy.

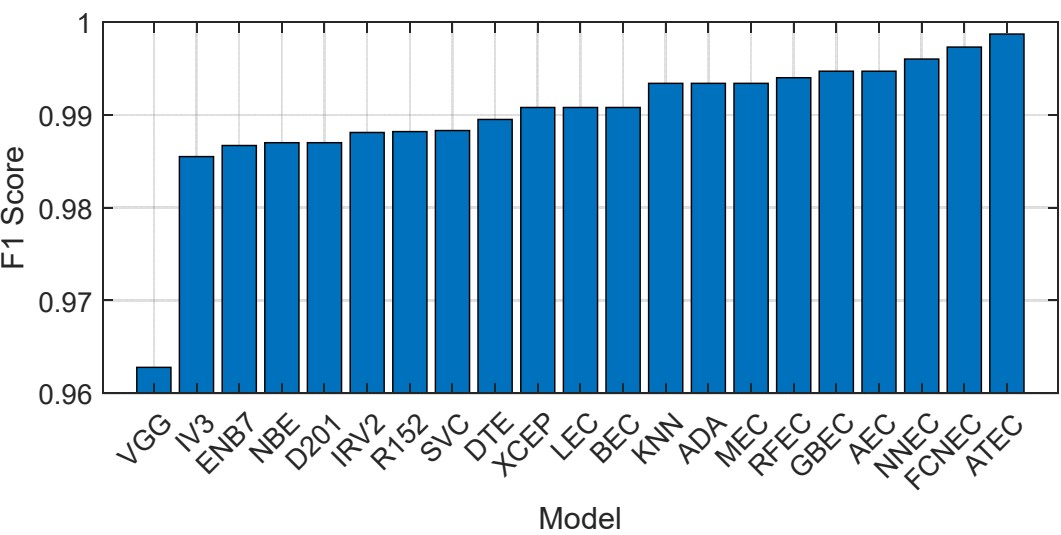

**Figure 18.** Prediction F1 Score.

Figures 19 and 20 summarize the total parameters and inference time of the presented models. The depth wise separable convolution in the Xception feature extractor allows for a smaller number of parameters, thereby reducing overfitting. The dense block in the DenseNet feature extractor allows for better gradient propagation and feature sharing. The Voting Ensemble CNNs models combined the four Deep CNN classification models where the classification heads took up the greatest number of parameters. The Voting models used the trained base models for inference and hence took less time for inference. The presented Deep and Shallow Meta-Learning Ensemble CNNs models employed meta-learners after the convolutional layers and hence contained a smaller number of parameters. The Attentive Ensemble took the longest time for inference due to the computationally heavy attention mechanism.

The trained models can be directly served on a cloud service such as Vertex AI on Google Cloud. The chest CT scan images obtained can be sent to the model endpoint using REST API provided by the cloud service provider. The model predictions can be sent back to the end user, thus aiding the medical diagnosis. The images can be sent in batches or in streams for offline or online predictions, respectively, depending on the application needs.

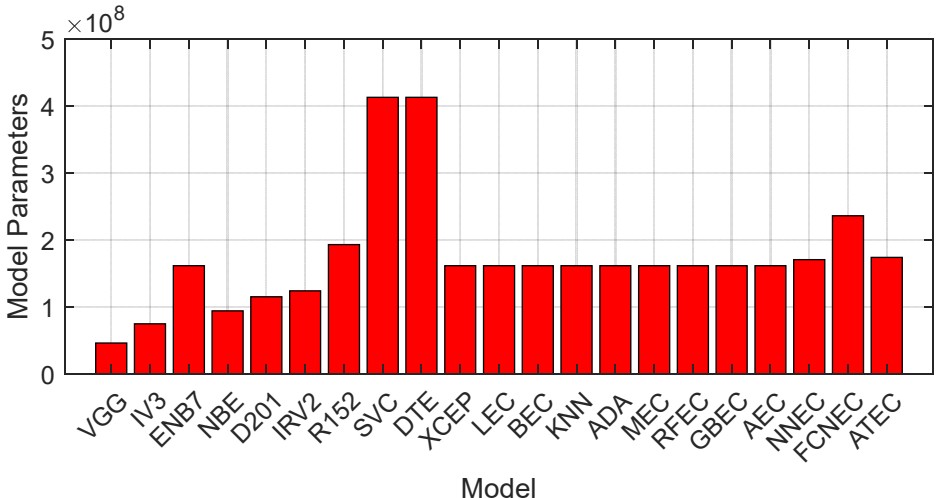

**Figure 19.** Ensemble Model Parameters.

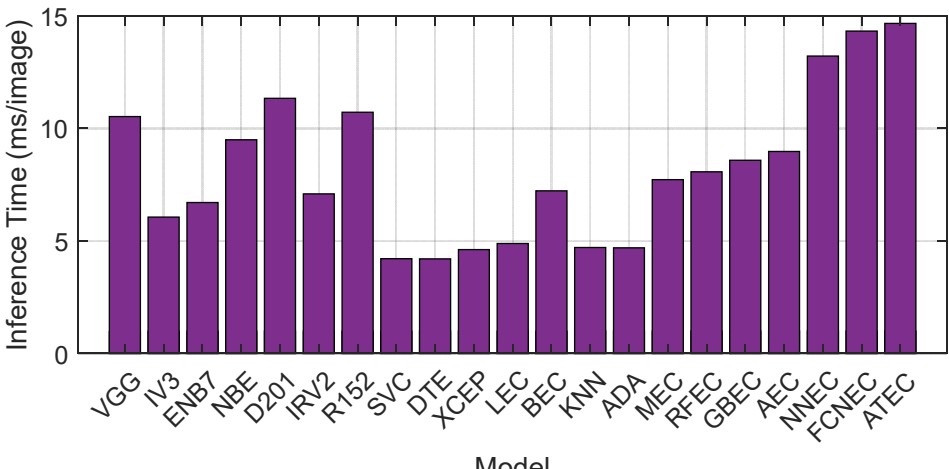

**Figure 20.** Inference Time of Ensemble Models.

## 5. Conclusions

This paper presented Deep Ensemble Learning architectures to classify infected chest CT scan images as COVID-19, as Viral Pneumonia, or as Normal. Popular Deep CNN architectures were trained to classify these images using five-fold cross-validation and the top four best performing models (ResNet-152-v2, Densenet-201, Xception, and EfficientNet-B7) on the test database were employed as trained base models for Ensemble learning. Voting Ensemble CNNs models, Shallow meta-learning Ensemble CNNs models, and Deep meta-learning Ensemble CNNs models were then presented in this study. The Voting Ensemble CNNs model utilized the final prediction probabilities of the four trained CNN models. The Shallow meta-learning Ensemble CNNs models utilized the feature vectors extracted by the Deep CNNs to train classical machine learning models. The Deep meta-learning Ensemble CNNs models utilized the feature maps extracted by the Deep CNNs to train deep meta-learners (NN or FCN or Attention). The presented Deep meta-learning Ensemble CNNs models outperformed the Deep CNNs models, Conventional Ensemble CNNs models, and Shallow meta-learning Ensemble CNNs models achieving state-of-the-art results on multiple evaluation metrics. This study also proposed a novel Attentive Ensemble CNNs model that learns the relative importance between the feature embeddings, thereby allowing the model to focus on relevant feature embeddings while making the prediction. The improved performance of the Deep Ensemble Learning models can be attributed to the reduced bias and variance obtained as a result of increased model complexity and the synergized deep feature maps.

Future developments of the proposed approach include using 3D CT scans to train 3D CNN Ensemble models. Vision Transformers can also be tested as base learners by replacing Deep CNNs. The proposed approach can also be used for CT scan segmentation by training on well-annotated CT scans. This can enable the radiologists to locate the exact area of the infection. Activation maps of the Deep CNN base learners can be visualized to identify the part of the CT scan image that proved to be relevant for the particular classification. The proposed classification approach can be extended to more classes by increasing the variety and volume of the database.

**Author Contributions:** Conceptualization, J.B.T. and S.K.V.; methodology, J.B.T., S.K.V., S.M.S. and A.A.-J.; software, J.B.T.; validation, J.B.T., S.K.V., S.M.S. and A.A.-J.; formal analysis, J.B.T., S.K.V., S.M.S. and A.A.-J.; data curation, J.B.T. and S.K.V.; writing—original draft preparation, J.B.T.; writing—review and editing, J.B.T., S.K.V., S.M.S. and A.A.-J.; Funding: A.A.-J.; All authors have read and agreed to the published version of the manuscript.

**Funding:** This research has no external funding.

**Data Availability Statement:** The code of the paper is available at: https://github.com/jibin-t-2k/Deep-Ensemble-CNNs-for-COVID-19-Chest-CT-Classification (accessed on 6 January 2023).

**Conflicts of Interest:** The authors declare no conflict of interest.

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
