# Peer review of "Deep Feature Meta-Learners Ensemble Models for COVID-19 CT Scan Classification"

_electronics, doi:10.3390/electronics12030684_

Round 1

Reviewer 1 Report

The manuscript “Deep Feature Meta-Learners Ensemble Models for Covid CT Scan Classification” presented by Thomas et al. aims at analyse different machine learning models in the task of CT scan classification, in order to recognise COVID versus viral lung infections.

The experiments are well conducted and the manuscript is well written, however I have some minor concerns about the structure of this paper, data representation and clinical considerations, listed ad follows:

1)      Affiliation 4 is the same as affiliation 3

2)      Line 47 - Reference 3 is about artificial intelligence applied to CT analysis, instead of a clinical trial on RT-PCR accuracy.

Also, that manuscript is from August 2020 and its 2 references regarding RT-PCR accuracy are earlier, both from China and on relatively small datasets (51 and 410 patients, respectively).

However, today RT-PCR performed on nasopharyngeal swab tests is still the gold standard for early diagnosis of COVID-19, due to better cost/time-effectiveness ratio and avoidance of unnecessary radiation to patients.

In addition to these considerations, the accuracy of CT diagnosis can be really helpful in determining whether a suspected lung infection is caused by SARS-Cov2 or another virus/pathogen.

Moreover the Authors should take into account that every dataset they used had perfomed COVID-19 diagnosis through RT-PCR.

I suggest to the Authors to rewrite this phrase includig more recent and reliable references on RT-PCR accuracy, but focusing mostly on aiding CT diagnosis, without proposing CT scans as an alternative to RT-PCR, since there were and there are many patients positive to COVID-19 without signs of lung infection.

3)      Lines 129-134 - I really suggest to the Authors to completely rework the structure of the paper accordingly with the journal's template: 1. Introduction; 2. Materials and Methods; 3. Results; 4. Discussion; 5. Conclusions.

Moreover, I suggest to include database information prior to algorithms descriptions; this could help the readers to better understand the proposed methods. Specifically, a well structured Materials and Method section should include: the database description, the "Deep CNN Architectures" section, the actual "Methodology" section and the "Experiment setup" sub-section.

4)      Line 215 – I suggest to modify the mention of references: "...as mentioned in previously published manuscripts [Refs.]" or "...as mentioned by AUTHOR et al. and AUTHOR et al. [Refs.]"

5)      Line 221 - Do you mean 224x224 pixel for every CT slice? In which step do you perform this resizing operation? Is this a resampling? If so, is this downsampling or upsampling? Why did yo choose this resolution? Which is the resolution of CT scans of the used database?

6)      Line 222 – Range 1,1 - What is this range about? Colors (black to white) in CT slices? Or width and height measures of every CT slice? Please clarify.

7)      Figure 2 and 7 - All images with stacked parallelepids: please invert the layer order of parallelepids of different colours to appear as a column. Since these images appear to be realised with word or powerpoint, you can use the "send back" function.

8)      Figure 2 caption - To better aid the readers, please add the description of which chapter do you describe the Ensemble Learning Models (e.g., "Average and Majority Ensemble CNNs are described in section 3.1").

9)      Line 309 - Reference 43 utilises the same datasets of the presented manuscript, adding only the dataset included in the following link: http://medicalsegmentation.com/covid19/

Even if it is obvious, I suggest to the Authors to specify that data are not duplicated (e.g., "removing duplicated datasets") or at least including the added dataset (i.e., the link mentioned above) as reference instead of the actual one.

10)   Line 322 - Please correct this paragraph, you have included the entire caption of figure 11 instead of mentioning the figure.

11)   Section 4.2 Experiment setup - I suggest to  the Authors to include more information about the scripting language used (I suppose it is Python) and/or include the code as supplementary material, if possible.

12)   Line 328 - How did you perform the split? Did you use a random choice algorithm or just a sequential splitting?

13)   Tables 2, 3, 4 and 5 - In order to help the readers to better understand these tables, I suggest to the Authors to move the Accuracy column as second or last column.

14)   Figure 13, 15 and 16 - I suggest to the Authors to modify the current image as a unique image, and to position the label on a corner, in order to resize avery single image to enhance readibility.

15)   Section 4.3.8 - Since the ROC images, apart for the figure 17, are very similar and not really helpful, I suggest to move at least the images in a supplementary material file, with larger and therefore more readible images.

16)   Lines 458-460 – I suggest to modify the mention of references as “as proposed by NAME [Ref], Name [Ref], ...”

17)   Table 6 - I suggest to include a column for classifying the models (e.g., CNN models, Voting ensemble, etc…)
Moreover, I wonder if could be helpful to calculate the inference average operation number (maybe dividing FLOPS by inference time), since I believe that inference time is strongly dependant on platform FLOPS capacity.

18)   Section 4.2 – As mentioned above, this should be a main sention and not a sub-section.

19)   Discussion and Conclusion - In the discussion and conclusions, the Authors should discuss the future applications of these models.

In particular, aiming at the clinical application of the radiologists' diagnostic aid, a clinical study should be conducted.

Furthermore, I suggest to the Authors to discuss the possible future developments.

The first may be to train these models to also predict other lung infections and diseases, including neoplasties, taking into account the limitation of a reduced number of available images.

Another development could be the visualization on images of characteristics recognized as determinants in the diagnosis, as proposed by Maftouni et al. in Figure 4 of Ref 43, as the radiologist would be more confident in accepting a diagnostic suggestion if highlighted on CT slices, rather than just as textual information.

20)   Line 541 – Please insert access date.

Author Response

please refer to the attached file as all the reviewer comment has been considered.

Reviewer 2 Report

The authors proposed proposes a deep ensemble learning to improve performance. This work is meaningful and some comments below may help authors to improve the presentation of this article.

1. The Introduction part can be simple and core clearly, you should write concise and explicit about your main contributions..

2. The proposed method in this paper is only used in the COVID dataset and has achieved good results. Please explain the generalization and robust performance of the proposed framework.

3. The authors propose a multiple method ensemble, which is derived from the fusion of multiple algorithms. Please explain the theoretical computational complexity of the framework.

4. The authors use CNN for global feature extraction and texture feature extraction, more details can be added.

5. What is the initialized parameters for meta-learning, and how to choose best?

6. Some core equations of proposed method are necessary. The optimize flow may display by pseudo-code.

7. Why choose meta optimization, not including grid search, random search, genetic algorithm, paticle swarm optimization. How to ensure optimization is robustness.

8. In the experimental part of section 4, what is way to select a part of dataset for meta learning.

9. The proposed method only runs a dataset, the performances are better other approaches, however, how to verify the proposed method is robustness? This web gives 6752 samples and 617,775 slices. Chest CT images are deposited into the China National Center for Bioinformation at the website ( http://ncov-ai.big.ac.cn/download?lang=en).

10. In Section 3, the author only verified the value of the best performance obtained in the experiment {27,...,34}, please conduct comparative experiments with more values.

11. The gender or age is a key factor, considering it may have new gains.

12. The newest COVID-CT approaches may add this work as compared learning methods. For example, references methods can be included.

13. G-mean can be measured imbalance dataset, which is a metric in this work.

14. Descriptions of COVID19 biology significance combing deep learning methods are given in the section of Results or Discussion.

Author Response

(The authors gave the same response as above.)

Round 2

Reviewer 1 Report

The Authors addressed all important comment and the majority of less important ones.

Therefore I suggest to the Editor to accept this manuscript in the present form.

Author Response

Reviewer 1 thank you

Reviewer 2 Report

More metrics can measure performance of imbalanced dataset effectively, such as G-mean.

Author Response

Response to reviewer 2
